# Herb Robert’s Gift against Human Diseases: Anticancer and Antimicrobial Activity of *Geranium robertianum* L.

**DOI:** 10.3390/pharmaceutics15051561

**Published:** 2023-05-22

**Authors:** Łukasz Świątek, Inga Wasilewska, Anastazja Boguszewska, Agnieszka Grzegorczyk, Jakub Rezmer, Barbara Rajtar, Małgorzata Polz-Dacewicz, Elwira Sieniawska

**Affiliations:** 1Department of Virology with Viral Diagnostics Laboratory, Medical University of Lublin, Chodźki 1, 20-093 Lublin, Poland; anastazja.boguszewska@umlub.pl (A.B.); barbara.rajtar@umlub.pl (B.R.); malgorzata.polz-dacewicz@umlub.pl (M.P.-D.); 2Student Research Group, Department of Virology with Viral Diagnostics Laboratory, Medical University of Lublin, Chodźki 1, 20-093 Lublin, Poland; wasilewskainga@gmail.com (I.W.); jakubrezmer1@gmail.com (J.R.); 3Chair and Department of Pharmaceutical Microbiology, Medical University of Lublin, Chodźki 1, 20-093 Lublin, Poland; agnieszka.grzegorczyk@umlub.pl; 4Department of Natural Products Chemistry, Medical University of Lublin, Chodźki 1, 20-093 Lublin, Poland

**Keywords:** *Geranium robertianum*, antiherpesviral, antimicrobial, anticancer

## Abstract

*Geranium robertianum* L. is a widely distributed plant used as a traditional herbal medicine, but the knowledge of its biological properties still needs to be improved. Thus, the purpose of this presented research was to assess the phytochemical profile of extracts from aerial parts of *G*. *robertianum*, commercially available in Poland and to study their anticancer potential and antimicrobial properties, including the antiviral, antibacterial, and antifungal effects. Additionally, the bioactivity of fractions obtained from the hexane and ethyl acetate extract was analyzed. The phytochemical analysis revealed the presence of organic and phenolic acids, hydrolysable tannins (gallo- and ellagitannins), and flavonoids. Significant anticancer activity was found for *G. robertianum* hexane extract (GrH) and ethyl acetate extract (GrEA) with an SI (selectivity index) between 2.02 and 4.39. GrH and GrEA inhibited the development of HHV-1-induced cytopathic effect (CPE) in virus-infected cells and decreased the viral load by 0.52 log and 1.42 log, respectively. Among the analyzed fractions, only those obtained from GrEA showed the ability to decrease the CPE and reduce the viral load. The extracts and fractions from *G. robertianum* showed a versatile effect on the panel of bacteria and fungi. The highest activity was observed for fraction GrEA4 against Gram-positive bacteria, including *Micrococcus luteus* ATCC 10240 (MIC 8 μg/mL), *Staphylococcus epidermidis* ATCC 12228 (MIC 16 μg/mL), *Staphylococcus aureus* ATCC 43300 (MIC 125 μg/mL), *Enterococcus faecalis* ATCC 29212 (MIC 125 μg/mL), and *Bacillus subtilis* ATCC 6633 (MIC 125 μg/mL). The observed antibacterial effect may justify the traditional use of *G. robertianum* to treat hard-to-heal wounds.

## 1. Introduction

*Geranium robertianum* L. is a plant widely distributed across a wide area from Europe to China to Japan; it is also found in Africa (southward as far as Uganda), the Atlantic seaboard of North America, and the temperate regions of South America [1]. *G. robertianum* L., commonly known as herb Robert or red robin, has long been used in folk and herbal medicine in many countries to treat digestive system disorders. Other common names of this plant include dragon’s blood, stork’s bill, and wild crane’s bill [1,2]. Its antiinflammatory, hemostatic, antidiabetic, antibacterial, antiallergic, anticancer, and diuretic properties have also been described [2]. The inhabitants of the high mountainous regions of Montenegro have indicated the above-ground parts of the *G. robertianum* as applicable to diarrhea, stomach inflammation, diseases of the gallbladder, kidneys, bladder, and ureters and externally to hard-to-heal wounds and mild rashes [3]. Even though *G. robertianum* L. has been reported to be commonly used in traditional folk medicine to treat various diseases, the actual biological activity is surprisingly understudied. So far, studies have confirmed the antibacterial activity (against a very limited panel of bacteria), as well as the antioxidant, antiinflammatory, and antihyperglycemic activities [2]. Moreover, the cytotoxic influence on cancer cells, including MCF-7 (breast adenocarcinoma), NCI-H460 (non-small cell lung cancer), HeLa (cervical carcinoma), and HepG2 (hepatocellular carcinoma), has been described [4].

Previous studies of the phytochemical composition of *G. robertianum* L. have indicated the presence of phenolic compounds, such as tannins, phenolic acids, and flavonoids. Ellagitannins are the most frequently isolated tannins, the main being geraniin. In addition, the presence of proanthocyanidins has been reported. Phenolic acids are represented chiefly by gallic, ellagic, ferulic, caffeic, and chlorogenic acids. Flavonoids are abundant in this plant, including quercetin and kaempferol, as aglycones or in glycosidic combinations. Non-phenolic compounds have also been described, such as lectins, saponins, and alkaloids [2].

Human herpesvirus type-1 (HHV-1; (herpes simplex virus type-1, HSV-1)) belongs to the Alphaherpesvirinae subfamily within the Herpesviridae family, together with human herpesvirus type-2 (HHV-2; HSV-2) and varicella zoster virus (VZV; HHV-3). A common feature of these viruses is the ability to infect and replicate in the epithelial tissues, with a subsequent invasion of the nervous system and establishment of lifelong latency. Reactivation may be asymptomatic or lead to recurrences of oral, labial, or ocular lesions, infrequently more severe sequelae, such as herpetic encephalitis [5,6]. Several treatment options for HHV-1 infections are available, but rapid development of drug resistance resulting in the loss of effectiveness is observed. Thus, it is imperative to look for alternative drugs with antiviral properties. Natural products (NP) have been used for centuries to treat diseases, including infections. Notably, many NP medicinal plants and their bioactive constituents have proven to exert antiherpesviral properties [6]. Among them, several species belonging to the Geranium genus were found to exert antiviral potential, inhibiting not only herpesviruses [7] but also the influenza virus [8] and hepatitis B virus (HBV) [9]. We have previously reported that the methanolic extract of *Geranium pyrenaicum*’s aerial parts showed significant antiviral activity against HHV-1, inhibiting the virus-induced cytopathic effect and reducing the virus infectious titer and viral load [10]. This study drove our attention to other plants belonging to the Geranium genus, especially *G. robertianum*, which is commonly found growing wild and cultivated in Poland. There are no reports on the antiviral studies of *G. robertianum* in the literature. That is why we have attempted to fill this gap by evaluating the antiherpesviral activity of this plant.

Both traditional use and the literature data point to the anticancer potential of *G. robertianum* [2,11], but this activity has not been previously assessed regarding cancer belonging to the head and neck cancers (HNCs) or colon cancer. Despite the previously mentioned traditional use of *G. robertianum* in treating wounds and skin rashes, data on the antibacterial and antifungal activity are still limited, focusing mainly on the distillated essential oil [12]. To assess the antibacterial potential of *G. robertianum,* we have selected a panel of Gram-positive and Gram-negative bacteria and selected *Candida* spp.

Considering the information presented herein, we have decided to enrich the scientific knowledge of *G. robertianum* by performing a phytochemical analysis of extracts from aerial parts of this plant obtained with four different polarity solvents, using chromatographic separation with mass spectrometry detection (LC-ESI-QTOF-MS/MS). Initial screening showed that the hexane and ethyl acetate extracts exerted desired biological activity; thus, those extracts were further separated into eight and six fractions, respectively. Subsequently, the cytotoxicity against non-cancerous and cancer cells was assessed and antimicrobial properties, including antiviral, antibacterial, and antifungal activity, were analyzed.

## 2. Materials and Methods

### 2.1. Plant Material

The dried herb of *Geranium robertianum* produced by DARY NATURY sp. z o.o. Koryciny 73, 17-315 Grodzisk, Poland was used. DARY NATURY is a certified manufacturer and distributor of herbal products. *G. robertianum* herb was certified as an organic product (Certificate PL-EKO-01-001493) according to Regulation (EU) 2018/848 of the European Parliament and of the Council of 30 May 2018 on organic production and labeling of organic products. The manufacturer ensured that the plant material was of high quality and was collected at the appropriate stage of development. Before the extraction, the herb was ground in a mechanical grinder and sieved through a sieve with a mesh size of 1 mm.

### 2.2. Preparation of Extracts for Screening

Appropriately powdered and sieved *G. robertianum* herb was weighed in 10.0 g for each tested extract and placed in four separate flat-bottomed 250 mL flasks. Then, 200 mL of methanol was added to the first flask containing plant material, 200 mL of ethyl acetate to the second flask, and 200 mL of hexane to the third flask. The mixtures prepared this way were closed with stoppers and maceration was carried out for 24 h at room temperature. After this time, the extracts were filtered through filter paper; then, the solvents were evaporated under reduced pressure until dryness was reached. The remaining 10.0 g of plant material was prepared by an infusion method; the plant material was poured with 200 mL of boiling water and left for 15 min. After this, the extract was filtered, cooled, frozen, and lyophilized. All extracts were stored at 4 °C until analysis.

### 2.3. Compound Identification

The chemical composition of *G. robertianum* extracts was determined with chromatographic separation and mass spectrometry detection (Agilent 1200 Infinity HPLC coupled to Agilent 6530B QTOF, Agilent Technologies, Santa Clara, CA, USA). The C18 Gemini^®^ chromatographic column (3 µm i.d. with TMS endcapping, 110 Å, 100 × 2 mm) protected by a guard column (Phenomenex Inc., Torrance, CA, USA) was used to separate compounds with the application of the gradient elution in the solvent composed of 0.1% formic acid in water (*v*/*v*) (solvent A) and 0.1% formic acid in acetonitrile (*v*/*v*) (solvent B). The flow rate was maintained at 0.2 mL/min during the following program: 0–60% B for 45 min; next, 60–95% B for 1 min; then, 95% B for 4 min, operating at 20 °C. Eluted compounds were ionized in a Dual Agilent Jet Stream spray source (ESI) connected to an N_2_ generator and analyzed in negative ions mode. The nebulizer pressure was set at 40 psig, whereas the capillary voltage was 4000 V. The sheath gas temp and flow were 325 °C and 12 L/min, respectively, while the drying gas temp and flow were 275 °C and 10 L/min, respectively. The MS and MS/MS acquisition was performed in the *m*/*z* range from 50 to 1500, maintaining the skimmer at 65 V, the fragmentor at 140 V, and a collision energy of 10 and 30 eV. PubChem (an open chemistry database (https://pubchem.ncbi.nlm.nih.gov/ (accessed on 1 March 1 2023)) and the literature data were used to compare empirically obtained fragmentation patterns with published mass spectra.

### 2.4. Preparation of Hexane and Ethyl Acetate Fractions

The hexane (3.89 g) and ethyl acetate (7.04 g) extracts were obtained from 400 g and 500 g of grounded herb, respectively, in the same way as described in Section 2.2. The extracts dissolved in a small volume of respective solvents were mixed with silica gel in the ratio 1: 10 and the silica gel was let to dry. Then, the silica gel with the adsorbed extract was put into a Büchner funnel and rinsed under reduced pressure with portions of solvents of increasing polarity. For hexane extract, the following mixtures were used: hexane-diethyl ether 9 + 1, 400 mL; hexane-diethyl ether 8 + 2, 200 mL; hexane-diethyl ether 6 + 4, 200 mL; hexane-diethyl ether 4 + 6, 200 mL; hexane-diethyl ether 2 + 8, 200 mL; diethyl ether, 200 mL; ethyl acetate, 200 mL; acetone, 200 mL. This yielded fractions GrH1–GrH8. For the ethyl acetate extract, the following mixtures were used: hexane-ethyl acetate 8 + 2, 400 mL; hexane-ethyl acetate 6 + 4, 400 mL; hexane-ethyl acetate 4 + 6, 400 mL; hexane-ethyl acetate 2 + 8, 400 mL; ethyl acetate, 600 mL; acetone, 600 mL. This yielded fractions GrEA1–GrEA6. Fractions were evaporated to dryness under reduced pressure and stored before analysis. The yields of obtained fractions can be found in Appendix A.

### 2.5. Cytotoxicity and Antiviral Activity

Cell lines used in this research included a non-cancerous VERO (ATCC (American Type Culture Collection), CCL-81) cell line and cancer cell lines: FaDu (ATCC, HTB-43), Detroit 562 (ATCC, CCL-138), and RKO (ATCC, CRL-2577). The human herpesvirus type-1 (HHV-1, ATCC, VR-260) was cultured in VERO cells, titrated using end-point titration assay and stored at −76 °C. The HHV-1 infectious titer was 5.5 ± 0.25 log_10_CCID_50_/mL (CCID_50_—50% cell culture infectious dose). Experiments were conducted under aseptic conditions in the BSL-2 laboratory. Incubation was carried out in a 5% CO_2_ humidified atmosphere at 37 °C (CO_2_ incubator, Panasonic Healthcare Co., Tokyo, Japan). Cellular morphology was monitored using an inverted microscope (CKX41, Olympus Corporation, Tokyo, Japan) equipped with a camera (Moticam 3+, Motic, Hong Kong, China) and software for image documentation (Motic Images Plus 2.0, Motic). Stock solutions of extracts were prepared by dissolving (50 mg/mL) the extracts in cell culture grade DMSO. Stock solutions of extracts were stored frozen (−23 °C) until used. The details of cell culturing, materials used, and experiments are included in the Appendix A.

#### 2.5.1. Evaluation of Cytotoxicity

The cytotoxicity of crude extracts (aqueous, methanolic, ethyl acetate, and hexane), as well as fractions obtained from ethyl acetate extract (GrEA1–GrEA6) and hexane extract (GrH1–GrH8), was evaluated using a previously described [10] MTT assay. Briefly, the cellular monolayers of appropriate cell lines in 96-well flat-bottomed plates were treated with serial dilutions of stock extracts (1000–1 µg/mL) or fractions (500–0.5 µg/mL) in cell media for 72 h. After the incubation, the MTT assay was performed as described in the Appendix A. The results were evaluated using GraphPad Prism (version 7.04, GraphPad Software, Inc., La Jolla, CA, USA); the CC_50_ (concentration decreasing the viability by 50%) values were calculated from dose–response curves (non-linear regression model).

#### 2.5.2. Antiviral Activity

The antiviral activity was based on assessing the influence of *G. robertianum* extracts and fractions on the development of HHV-1-induced cytopathic effect (CPE) in infected VERO cells and the semi-quantitative analysis of the viral load with the real-time PCR. Acyclovir (60 µg/mL) was used as a reference antiviral drug [10].

The monolayer of VERO cells in 48-well plates was infected with HHV-1 (100-fold CCID_50_/mL) in cell media for 1 h, leaving at least two uninfected wells (cell control). After infection, the media were removed, the cells were washed with PBS, and the non-toxic concentrations of *G. robertianum* extracts or fractions in the cell media were added. The non-infected VERO cells and the non-treated infected cells (virus control) wells were supplemented with media containing 2% FBS. The plates were incubated until the cytopathic effect (CPE) was observed in the virus control (approx. 72 h). Afterwards, the influence of tested samples on the development of CPE was compared with the CPE in the virus control; the results were documented. Finally, the plates were thrice frozen (−72 °C) and thawed and samples were collected and stored at −72 °C until used for DNA isolation.

The viral DNA was isolated from the collected samples using a commercially available kit (QIAamp DNA Mini Kit, QIAGEN GmbH, Hilden, Germany) following the manufacturer’s instructions. The real-time PCR amplification was carried out with SybrAdvantage qPCR Premix (Takara Bio Inc., Kusatsu, Shiga Prefecture, Japan) and primers (UL54F—5′ CGCCAAGAAAATTTCATCGAG 3′, UL54R—5′ ACATCTTGCACCACGCCAG 3′), using the CFX96 thermal cycler (Bio-Rad Laboratories, Inc., Hercules, CA, USA). The amplification parameters can be found in the Appendix A. The viral load of HHV-1 in samples treated with *G. robertianum* extracts and fractions was assessed in relation to the virus control based on the relative quantity (ΔCq) method using CFX Manager™ Dx Software (Version: 3.1.3090.1022; Bio-Rad Laboratories).

### 2.6. Antibacterial and Antifungal Activity

The various extracts (GrM, GrAq, GrH, and GrEA) and fractions from two extracts (GrEA1–6 and GrH1–8) from *G. robertianum* were tested against a panel of standard microorganisms belonging to the American Type Culture Collection (ATCC) including Gram-positive bacteria: *Staphylococcus aureus* ATCC 29213, *Staphylococcus aureus* ATCC 6538P, and *Staphylococcus aureus* ATCC 25923; methicillin sensitive strains: *Staphylococcus aureus* ATCC 43300 and *Staphylococcus aureus* ATCC BAA1707; methicillin resistant strains: *Staphylococcus epidermidis* ATCC 12228, *Enterococcus faecalis* ATCC 29212, *Micrococcus luteus* ATCC 10240, *Bacillus subtilis* ATCC 6633, and *Bacillus cereus* ATCC 10876; Gram-negative bacteria: *Salmonella* Typhimurium ATCC 14028, *Proteus mirabilis* ATCC 12453, *Bordetella bronchiseptica* ATCC 4617, *Escherichia coli* ATCC 25922, and *Pseudomonas aeruginosa* ATCC 27853; yeasts: *Candida albicans* ATCC 2091, *Candida albicans* ATCC 10231, *Candida auris* CDC B11903, *Candida glabrata* ATCC 90030, *Candida parapsilosis* ATCC 22019, *Candida krusei* ATCC 14243, *Candida lusitaniae* ATCC 34449, and *Candida tropicalis* ATCC 1369. The broth microdilution method employed for the determination of antimicrobial activities of the extracts and all fractions was conducted according to the recommendations of EUCAST (the European Committee on Antimicrobial Susceptibility Testing) guidelines [13], as previously described [14]. Bacterial species were cultured at 35 °C for 24 h on Mueller–Hinton agar (MHA) and Mueller–Hinton broth (MHB). The yeast species were also cultured at 30 °C for 24 h on RPMI 1640 agar and RPMI 1640 broth. Bacteria and yeast were suspended in sterile saline to obtain an inoculum of 0.5 McFarland, corresponding to 1.5 × 10^8^ CFU (colony forming units) mg/mL for bacteria and 5 × 10^6^ CFU mg/mL for yeast. Minimum inhibitory concentrations (MICs) were determined by a serial dilution method in 96-well polystyrene microtiter plates against twenty-three in vitro cultured microorganisms. This method also allowed the determination of the minimum bactericidal concentration (MBC) or the minimum fungicidal concentration (MFC) of each extract and each fraction against these microorganisms. The extracts and fractions were dissolved in DMSO to obtain a final concentration of 100 mg/mL. The final concentrations tested were 8, 4, 2, 1, 0.5, 0.25, 0.125, 0.06, 0.03, 0.016, 0.008, and 0.004 mg/mL. Appropriate DMSO, growth, and sterile controls were carried out. The standard antimicrobial agents (fluconazole (0.06–16 µg/mL), ciprofloxacin (0.015–16 µg/mL), and vancomycin (0.06–16 µg/mL)) were used as antimicrobial substances active against yeasts, Gram-negative, and Gram-positive bacteria, respectively. The experiments were performed in triplicate.

## 3. Results

### 3.1. Phytochemical Profile of G. robertianum Extracts

The phytochemical analysis revealed the presence of secondary metabolites belonging to organic and phenolic acids, hydrolysable tannins (gallo- and ellagitannins), flavonoids, and fatty compounds. A detailed composition of all extracts is provided in Appendix A. The identification was based on the PubChem database and the literature sources [4,10,15,16,17]. Detected phenolic molecules occurred mainly in glycosylated forms. Among the phenolic acids, coumaroyl-, feruloyl-, and caffeoyl-quinic acids were detected. Gallic and ellagic acids were basic structures of tannins identified. Gallic acid gave rise to gallotannins (tetragalloylglucose, pentagalloylglucose) and ellagic acid, which in turn contributed to ellagitannins of several types. Corilagin represented galloyl and hexahydroxydiphenoyl (HHDP) group-based molecules and pedunculagin constituted from di-HHDP-glucose, while geraniin was based on the dehydrohexahydroxydiphenoyl (DHHDP) group. Castalin/vescalin originated from the C–C-connected trimer of the galloyl group. Galloyl groups were also conjugated to phenolic acids (galloylquinic acids, galloylchlorogenic acid) and glucose (galloylglucose, gallic acid O-(6-galloylglucoside)). The other derivatives of gallic acid were gallacetophenone, brevifolin, brevifolincarboxylic acid, and methyl brevifolincarboxylate. Flavonoids were represented by quercetin and kaempferol-based structures. Numerous non-polar molecules were tentatively assigned as fatty compounds or remained unidentified.

Many of the identified compounds consisted of a group of gallic acid and HHDP, hence the elimination of -galloyl (−152 Da), gallic acid (−170 Da), -galloylglucose (−332 Da), HHDP (−302 Da), or HHDP-glucose (−482 Da) groups visible in their spectra. The classification of the compounds into derivatives, based on ellagic acid or quercetin, was made based on fragmentation ions typical for these structures found in the MS/MS spectra. Fragmentation ions with *m*/*z* 283 [MH–H_2_O]−, 229 [MH–CO_2_–CO]−, 201 [MH–CO_2_–CO–CO]−, formed from a molecular ion with *m*/*z* 301, indicated the presence of molecules structurally related to ellagic acid and were distinguished from quercetin-based structures. Corilagin, with a molecular ion at *m*/*z* 633, gave fragmentation ions at *m*/*z* 463 [MH-170]- and 301 [MH-170-162]−, which were consistent with the loss of a gallic acid molecule (−170 Da) and one glucose molecule (−162 Da) bound to the HHDP group (302 Da). A similar fragmentation was observed for geranin, with the molecular ion at *m*/*z* 951 and fragmentation ions at *m*/*z* 463 and 301. Fragmentation ions at *m*/*z* 933 and 915 matched the successive elimination of the water molecule [MH–H_2_O]−. A compound with a molecular ion at *m*/*z* 783 and two fragment ions at *m*/*z* 481 [MH-302]− and 301 [MH-482]−, corresponding to the loss of HHDP and HHDP-glucose molecules, was identified as pedunculagin. Quercetin and kaempferol-based molecules were identified due to the presence of ions at *m*/*z* 301 and *m*/*z* 285 for quercetin and kaempferol, respectively. Quercetin yielded daughter ions at *m*/*z* 179 and 151, while kaempferol at *m*/*z* 151.

### 3.2. The Evaluation of Cytotoxicity towards Non-Cancerous and Cancer Cells

The evaluation of *G. robertianum* cytotoxicity was conducted using a microculture tetrazolium assay (MTT), as previously described [10], against non-cancerous VERO cells and cancer cell lines originating from pharyngeal cancers (FaDu and Detroit 562) and colon carcinoma (RKO). The highest toxicity towards VERO cells was observed for the *G. robertianum* methanolic extract (GrM) with CC_50_ (50% cytotoxic concentration) of 187.17 µg/mL (Table 1). The evaluation of the anticancer potential of GrM revealed that significant selectivity was present only against FaDu cells, with an SI of 1.61. The lowest cytotoxicity against all tested cell lines was noticed for the *G. robertianum* aqueous extract (GrAq) with selectivity only towards FaDu (SI 2.15). Significant anticancer potential was found for *G. robertianum* hexane extract (GrH) and ethyl acetate extract (GrEA) with SI 2.02 and 4.39, respectively. The dose–response curves of GrH against all cancer cells showed high similarity, confirming selective cytotoxic influence (Figure 1). Furthermore, the cytotoxicity of GrEA against FaDu and RKO was almost identical, whereas, against Detroit 562, lower cytotoxicity was observed.

Since GrH and GrEA showed promising anticancer potential, a fractionation was performed to evaluate whether this would lead to obtaining a fraction with higher activity. In the fractionation of GrH, subsequently received fractions showed increasing cytotoxicity towards cancer and non-cancerous cells (Table 2). However, an interesting exception was observed for RKO cells, where, from GrH1 to GrH6, an increase of cytotoxicity was observed ranging from CC_50_ of 238.5 to 29.59 µg/mL. In contrast, fractions GrH7 and GrH8 were significantly less toxic than GrH6, with CC_50_ of 187.0 and 125.1 µg/mL, respectively. The dose–response analysis (Figure 2) confirmed that RKO was noticeably more resistant to GrH7 and GrH8 than other cell lines. In fact, anticancer selectivity of GrH7 and GrH8 was observed only towards FaDu cells. Among the fractions obtained from GrH, the highest anticancer selectivity towards all cancer cells was found for GrH3 (SI 2.81–2.9). For GrH4–GrH6, higher selectivity was observed towards FaDu and RKO (SI 2.01–3.22) compared with Detroit 562 (SI 1.28–1.46).

The fractionation of GrEA also provided fractions with increasing cytotoxicity (Figure 3), except the last obtained fraction, GrH6, which showed lower cytotoxicity than GrEA5. Surprisingly, the first fraction—GrEA1—despite showing relatively weak (VERO) to moderate cytotoxicity (cancer cells), showed the highest anticancer selectivity, with an SI between 1.71 and 3.5. Overall, fractions GrEA1–GrEA4 showed selective anticancer activity towards FaDu (SI 3.02–3.5) and RKO (SI 1.94–3.22) and, to a lesser extent, against Detroit 562 (SI 1.44–1.71). Fractions GrEA5 and GrEA6 selectively inhibited only the FaDu cells, with an SI of 2.43 and 1.9, respectively.

Noteworthy, based on the criteria of plant extract cytotoxicity proposed by the National Cancer Institute (NCI) [18] and found in the literature [19], high cytotoxic activity can be reported for samples showing CC_50_ < 20 μg/mL. Thus, cytotoxicity results obtained for GrH7 (CC_50_ 13.35 µg/mL), GrH8 (CC_50_ 18.51 µg/mL), GrEA3 (CC_50_ 19.86 µg/mL), GrEA4 (CC_50_ 14.22 µg/mL), and GrEA5 (CC_50_ 17.89 µg/mL) on FaDu cells point to high cytotoxic activity.

### 3.3. Assessment of Antiherpesviral Activity

The antiviral activity was tested against the human herpesvirus type-1 (HHV-1) replicating in the VERO cell line. The HHV-1-infected VERO cells were incubated with tested extracts in non-cytotoxic concentrations. As can be seen in Figure 4E, the HHV-1-induced cytopathic effect (CPE) was present in the virus control (VC, HHV-1 infected, untreated cells). Non-infected cells (cell control) are presented in Figure 4A. The GrM and GrAq did not significantly influence the formation of CPE. However, GrH (Figure 4B) and GrEA (Figure 4F) at 150 µg/mL noticeably decreased the CPE, suggesting an influence on the replication of the HHV-1. This effect appeared to be dose-dependent; at lower concentrations (125 and 100 µg/mL) of GrH (Figure 4C,D) and GrEA (Figure 4G,H), an increase in CPE intensity was observed, with more cell rounding and lysis and occasional signs of syncytia formation.

Fractions GrH1–GrH8 did not influence the formation of virus-induced CPE and detailed results are presented in the Appendix A). However, the fractionation of GrEA allowed us to obtain fractions showing promising antiviral activity (Appendix A). Despite a clear presentation of CPE in the samples treated with GrEA1 (Appendix A), GrEA2 (Appendix A), and GrEA3 (Appendix A), it was noticeably less advanced than in the VC (Appendix A). Higher inhibition was observed for GrEA4 (Appendix A), with more cells showing normal morphology. However, the highest CPE inhibition was observed for GrEA5 (Appendix A) and GrEA6 (Appendix A). Importantly, no influence on the CPE was observed when the lower concentrations of GrEA1–GrEA4 were tested. However, samples GrEA5 and GrEA6 showed a dose–response effect in the range of 25–15 μg/mL for GrEA5 (Appendix A) and 50–30 μg/mL for GrEA6 (Appendix A). Acyclovir, a reference antiviral drug, inhibited the CPE (Appendix A).

This presented research evaluated the direct influence of tested extracts and fractions on the replication of HHV-1 by measuring the viral load reduction using real-time PCR. Based on the amplification and SYBR green-based quantification of HHV-1 specific DNA fragments, we evaluated the relative amount of viral copies in DNA isolates obtained from the antiviral experiments described above. Each experiment included a virus control; the viral load reduction was calculated with reference to the appropriate VC isolate using the ΔCq method (Appendix A). An example of real-time PCR amplification of DNA isolates from HHV-1 infected VERO cells treated with fractions obtained from *G. robertianum* ethyl acetate extract and the acyclovir (60 μg/mL) is presented in Figure 5A. An analysis of stock virus isolate and its dilutions were also included to evaluate the method sensitivity. It was found that the GrM and GrAq extracts showed a marginal effect (Δlog < 0.5 log) on the HHV-1 replication (Figure 5B), which corresponded to the results of their influence on the HHV-1-induced CPE. Interestingly, the GrH, despite noticeable CPE reduction, showed low viral load reduction by 0.52 log at 150 μg/mL. The GrEA, at 150 μg/mL, reduced the viral load by 1.42 log. A dose–response activity was observed and lower concentrations, 125 and 100 μg/mL, showed a reduction of 0.69 and 0.15 log (Figure 5B). Acyclovir managed to reduce the viral load by more than 6 log. Fractions GrH1–GrH8 and GrEA1–GrEA3 did not significantly affect the viral load and a reduction by 0.67 log was observed for GrEA4 at 20 μg/mL. The highest inhibition of HHV-1 replication was found for GrEA5 and GrEA6, with Δlog of 1.77 and 1.51 log, respectively. Melt curve analysis showed a single melt peak (Figure 5C), confirming that the same amplicon was present in all tested DNA isolates (except for the negative control).

### 3.4. Antimicrobial Activity

The results of an antimicrobial analysis of four *G. robertianum* extracts (GrM, GrAq, GrH, and GrEA) are presented in Figure 6. It can be seen that the extracts showed versatile activity against tested reference bacteria (MIC from 0.03 to ≥ 8 mg/mL) and yeasts (MIC = 1–8 mg/mL). The most susceptible strain was *B. subtilis* ATCC 6633 (MIC = 0.03–0.06 mg/mL). Overall, only the Gram-positive bacteria showed sensitivity to *G. robertianum* extracts.

Interestingly, the GrH extract from *G. robertianum* showed the highest activity (MIC = 0.06–0.5 mg/mL) against all Gram-positive bacteria. The GrEA extract also showed potent activity (MIC = 0.03–1 mg/mL) towards most Gram-positive bacteria, except *Staphylococcus aureus* ATCC BAA1707 (MIC = 2 mg/mL), a strain that is resistant to methicillin. The GrM extract also showed activity (MIC = 0.06–2 mg/mL) against Gram-positive bacteria. The GrEA and GrM were the only extracts showing activity against most tested yeasts, with MIC values ranging from 1 to 8 mg/mL. *Candida glabrata*, *Candida parapsilosis*, *Candida krusei,* and *Candida tropicalis* were the most sensitive to GrEA and GrM extracts (MIC = 1 mg/mL), while *Candida albicans* ATCC 2091 and *Candida albicans* ATCC 10231 only to GrEA (MIC = 1 mg/mL). *Candida auris* was resistant to all tested *G. robertianum* extracts.

Antimicrobial substances are usually regarded as bactericidal or fungicidal if the MBC/MIC or MFC/MIC ratio is ≤4. If the MBC/MIC or MFC/MIC ratio is >4, antimicrobial substances are usually regarded as bacteriostatic or fungistatic [14]. Based on the presented results, it can be concluded that only GrAq extracts showed bactericidal activity (MBC/MIC = 1–4) against all Gram-positive bacteria and GrAq and GrH extracts showed fungicidal effects (MFC/MIC = 1–4) against all tested yeast. The other extracts had versatile bactericidal/bacteriostatic and fungicidal/fungistatic effects. The MBC/MIC values could not be determined for most Gram-negative and some Gram-positive bacteria because the MBC was >8 mg/mL.

The fractions from *G. robertianum* hexane extract demonstrated divergent activity against Gram-positive bacteria (Appendix A) with MIC from 0.008 to ≥ 8 mg/mL. At the same time, the Gram-negative bacteria were unaffected (MIC ≥ 8 mg/mL). Among yeasts, only the *Candida parapsilosis* was more sensitive to GrH3, GrH7, and GrH8 fractions (MIC = 0.5–1 mg/mL) than the GrH extract (MIC = 2 mg/mL). The most susceptible bacterial strains were *Staphylococcus aureus* ATCC 43300, *Staphylococcus epidermidis*, *Enterococcus faecalis*, *Bacillus subtilis*, and *Bacillus cereus* with MIC ranging from 0.008 to 4 mg/mL. The highest activity was found for the GrH8 fraction against *Bacillus cereus*, *Bacillus subtilis*, *Micrococcus luteus*, and *Staphylococcus epidermidis* ATCC 12228, with the MIC ranging from 0.008 to 0.06 mg/mL. Moreover, the effect of GrH8 towards *Bacillus subtilis* and *Micrococcus luteus* appeared to be bactericidal. Potent bacteriostatic activity against *Staphylococcus epidermidis* was observed for GrH4 (MIC = 0.06 mg/mL).

The fractions from *G. robertianum* ethyl acetate extract showed antibacterial activity only against Gram-positive species, with MIC ranging from 0.008 to 4 mg/mL (Appendix A). The GrEA4 was found to exert a higher antifungal effect than GrEA against *Candida auris*, *Candida krusei*, and *Candida lusitaniae*, with MIC between 0.25 and 0.5 mg/mL. Interestingly, *Candida auris* was the most resistant fungal species to *G. robertianum* crude extracts, including GrEA. The most susceptible bacterial strains to fractions from GrEA extract were *Micrococcus luteus*, *Staphylococcus aureus* ATCC 43300, *Staphylococcus epidermidis*, *Enterococcus faecalis*, *Bacillus subtilis*, and *Bacillus cereus* (MIC= 0.008–4 mg/mL). All the fractions from GrEA showed potent activity against *Micrococcus luteus* ATCC 10240, with MIC between 8 and 16 μg/mL, whereas fractions 3, 4, and 5 were effective against *Staphylococcus epidermidis* (MIC = 16–60 μg/mL).

MIC values for the reference antimicrobial substances were 1 μg/mL of fluconazole for *Candida albicans* ATCC 10231, 1 μg/mL of vancomycin for *Staphylococcus aureus* ATCC 29213, and 0.015 μg/mL of ciprofloxacin for *Escherichia coli* ATCC 25922.

## 4. Discussion

Although *G. robertianum* is used in folk medicine in several countries to treat cancer, its toxicity to cancer cells has rarely been assessed. In our studies, *G. robertianum* crude hexane and ethyl acetate extracts showed selective anticancer potential towards human pharyngeal cancer cells (FaDu, Detroit 562) and colon cancer cells (RKO). Noteworthy, the fractionation we performed allowed for obtaining fractions that showed significantly higher toxicity than the crude extracts, some of them, such as GrH7, GrH8, GrEA3, GrEA4, and GrEA5, against FaDu with CC_50_ below 20 μg/mL. This, according to previously mentioned references [18,19], indicates high cytotoxicity. Some bioactive compounds identified in *G. robertianum* may be responsible for the observed anticancer activity. Gallic and ellagic acids are the two primary phenolic acids in *G. robertianum* and are commonly found in the *Geraniaceae* species [20,21]. Both compounds have a documented inhibitory effect on carcinogenesis [22,23]. Geraniin has also been reported to inhibit the cell growth of nasopharyngeal cancer cells (C666-1). Interestingly, geraniin promoted apoptosis and increased the accumulation of reactive oxygen species (ROS) in the C666-1 cells [24]. Moreover, flavonoids, quercetin, and kaempferol, alongside their glycosides, have been described to show cytotoxicity to several cancer cell types through increasing ROS accumulation, induction of apoptosis, and interactions with cancer-related signaling pathways (PI3K/Akt, MAPK, and NF-κB) [25].

Graça et al. [11] evaluated the effect of *G. robertianum* extracts on the growth of four human cancer cell lines: MCF-7 (breast adenocarcinoma), NCI-H460 (non-small cell lung cancer), HeLa (cervical cancer), and HepG2 (hepatocellular carcinoma), using sulforhodamine B assay. The acetone extract showed the highest toxicity, while the aqueous extracts (infusion and decoction) showed moderate cytotoxicity towards HepG2 (GI_50_ 45.68 and 52.2 μg/mL, respectively) [11]. In contrast, Catarino et al. [15] reported that aqueous defatted (using n-hexane) decoction from the same plant showed no cytotoxic influence on HepG-2 up to 100 μg/mL. Interestingly, the hexane extract showed GI_50_ between 151 and 179 μg/mL against all cancer cell lines [11], whereas, in our research, the hexane extract showed higher toxicity, with CC_50_ between 98.91 and 117.57 μg/mL against FaDu, Detroit 562, and RKO. The toxicity of *G. robertianum* extracts on normal cells points to the low toxicity of aqueous, methanolic, and ethyl acetate extracts (CC_50_ > 200 µg/mL) and moderate toxicity of methanolic extract. Indeed, health risks, including toxicity or side effects after treatment with *G. robertianum* using designated therapeutic dosages, were not recorded [1].

Paun et al. [26] studied the cytotoxicity of aqueous and aqueous–alcoholic (50/50 *v*/*v*) extracts of *G. robertianum* purified and concentrated in micro- and ultrafiltration processes against normal monkey kidney (RM) cells and Hep-2 cells, which the authors described as a human epidermoid laryngeal carcinoma cell line [26]. However, this cell line was earlier reported to comprise cervical adenocarcinoma cells derived via HeLa cell-line contamination [27]. Thus, our report is the first one describing the anticancer activity of *G. robertianum* extracts against cell lines originating from head and neck cancers (HNC).

This study was primarily inspired by our previous work reporting the antiherpesviral activity of *Geranium pyrenaicum* Burm. f. methanolic extract [10]. Since other studies have also pointed to the antiviral activity of plants belonging to the Geranium genus [7,8,9], we concluded it would be interesting to see whether *G. robertianum* extracts also exert activity against viruses. We have observed that the hexane and ethyl acetate extracts noticeably inhibited the development of the HHV-1-induced cytopathic effect. However, subsequent viral load semi-quantification based on real-time PCR showed a reduction of only 0.52 log and 1.42 log by GrH and GrEA (both at 150 µg/mL) compared with the virus control, respectively, which points to low antiviral potential. Surprisingly, the fractionation of GrH resulted in the loss of the ability to inhibit CPE development. However, fractions obtained from GrEA, namely GrEA4, GrEA5, and GrEA6, inhibited the virus-induced CPE development and showed higher viral load reduction than the GrEA crude extract. Moreover, this activity was observed at significantly lower concentrations for fractions (20–50 µg/mL) than for the crude GrEA extract (150 µg/mL). Summarizing, the *G. robertianum* hexane and ethyl acetate extracts and GrEA4, GrEA5, and GrEA6 fractions showed in vitro low antiviral activity against HHV-1. Several phytochemicals that we identified in *G. robertianum* have been previously described as exerting antiviral activity. Geraniin was reported to inhibit HSV-2 and, to a smaller extent, the HHV-1 virus [28]. We also observed the presence of geraniin in the *Geranium pyrenaicum* Burm. f. methanolic extract inhibiting the replication of HHV-1 [10]. The ellagic acid inhibited human rhinoviruses HRV-2, -3, and -4, with higher activity than ribavirin used as a control antiviral drug [29]. Recently, ellagic acid was identified as a potential inhibitor of NS3 helicase of the Zika virus, an arbovirus belonging to the *Flaviviridae* family [30,31]. Cui et al. [32] reported that ellagic acid and gallic acid exert antiviral activity against the Ebola virus by inhibiting virus entry into the cell. Notably, ellagic acid and gallic acid were also found to inhibit HHV-1 replication in VERO cells [33]. Thus, it can be concluded that phenolic acids may show broad-spectrum antiviral activity against DNA and RNA viruses. Flavonoids, such as quercetin and kaempferol, have also been reported as potential antivirals [34,35,36]. Quercetin and its derivates inhibit flaviviruses, herpesviruses, orthomyxoviruses, and coronaviruses [34], whereas kaempferol and its glycosides exerted antiviral potential against herpesviruses, such as HHV-3 [35], HHV-5 (cytomegalovirus, CMV) [36], and BoHV-1 (bovine herpesvirus 1) [37], and several other viruses [38,39].

Noticeable antibacterial effects against Gram-positive bacteria, including reference strains of *Staphylococcus epidermidis* ATCC 12228, *Enterococcus faecalis* ATCC 29212, *Bacillus subtilis* ATCC 6633, *Bacillus cereus* ATCC 10876, and several reference strains of *Staphylococcus aureus,* were observed for hexane (MIC = 0.06–0.5 mg/mL) and ethyl acetate (MIC = 0.03–1 mg/mL) extracts. Interestingly, *Staphylococcus aureus* ATCC BAA1707, a reference strain resistant to methicillin, showed much higher sensitivity to hexane extract (MIC = 0.5 mg/mL) than to ethyl acetate extract (MIC = 2 mg/mL). The fractionation of those extracts allowed us to obtain fractions with much higher antibacterial activity. Regarding antifungal properties, the ethyl acetate extracts showed the most promising activity among crude extracts with MIC = 1 mg/mL against all fungal strains, except *Candida auris* CDC B11903. In this study, Gram-negative bacteria were not sensitive to extracts and tested fractions in contrast to Gram-positive bacteria; this effect may result from differences in cell-wall structure [40]. The activity of *G. robertianum* against Gram-positive bacteria may, to some extent, validate the use of this plant in the treatment of hard-to-heal wounds. The aqueous extract of *G. robertianum* from the entire plant exerted bactericidal activity against two closely related species of cariogenic bacteria, *Streptococcus mutans* and *Streptococcus sobrinus* [41]. The 70% water–ethanolic extract from the plant’s above-ground parts did not show a significant antimycobacterial effect [42]. Radulović et al. [12] reported that the essential oil from the underground parts of *G. robertianum* showed higher activity than the essential oil from the aerial parts. Our study is the first one assessing the antimicrobial potential of *G. robertianum* against a wide panel of bacteria, both Gram-positive and Gram-negative, and several species of *Candida*. Additionally, we also report low antiviral potential against the HHV-1 of this plant species.

## 5. Conclusions

*Geranium robertianum* is a valuable medicinal plant, with versatile biological activities, used in the traditional medicine of many countries. Herein, we would like to contribute to knowledge about this plant by describing the phytochemical composition of four extracts prepared using different solvents and reporting the results of anticancer and antimicrobial activity. The hexane and ethyl acetate extracts showed selective anticancer potential towards cells originating from pharyngeal and colon cancers. The ethyl acetate extract and selected fractions showed low antiviral potential against the human herpesvirus type-1. Moreover, the observed antibacterial potential, shown mainly against Gram-positive bacteria and selected *Candida* fungi, may validate the traditional use of *G. robertianum* in managing hard-to-heal wounds.

## Figures and Tables

**Figure 1 pharmaceutics-15-01561-f001:**
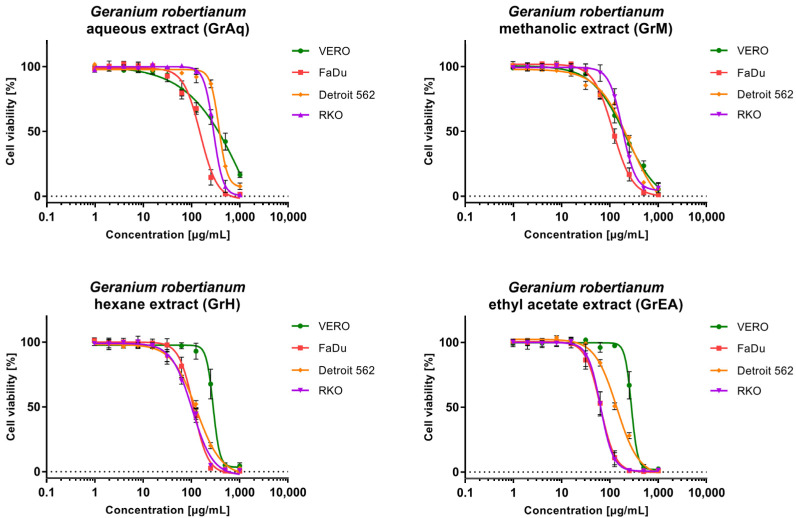
Dose–response effect of *G. robertianum* extracts on normal and cancer cells.

**Figure 2 pharmaceutics-15-01561-f002:**
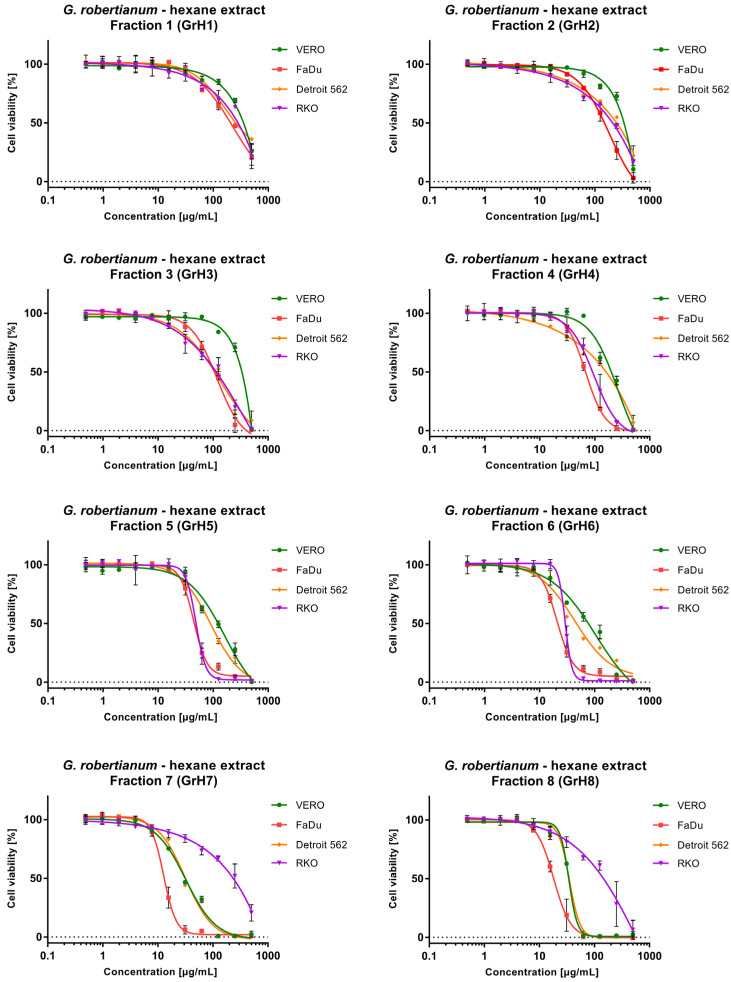
Cytotoxic effect of fractions isolated from *G. robertianum* hexane extract on a panel of cell lines.

**Figure 3 pharmaceutics-15-01561-f003:**
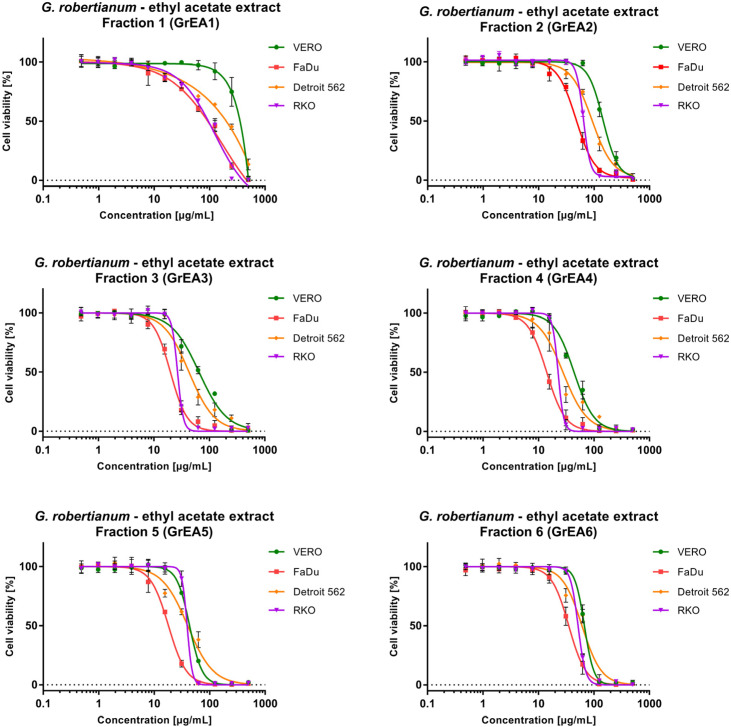
Cytotoxic effect of fractions isolated from *G. robertianum* ethyl acetate extract on a panel of cell lines.

**Figure 4 pharmaceutics-15-01561-f004:**
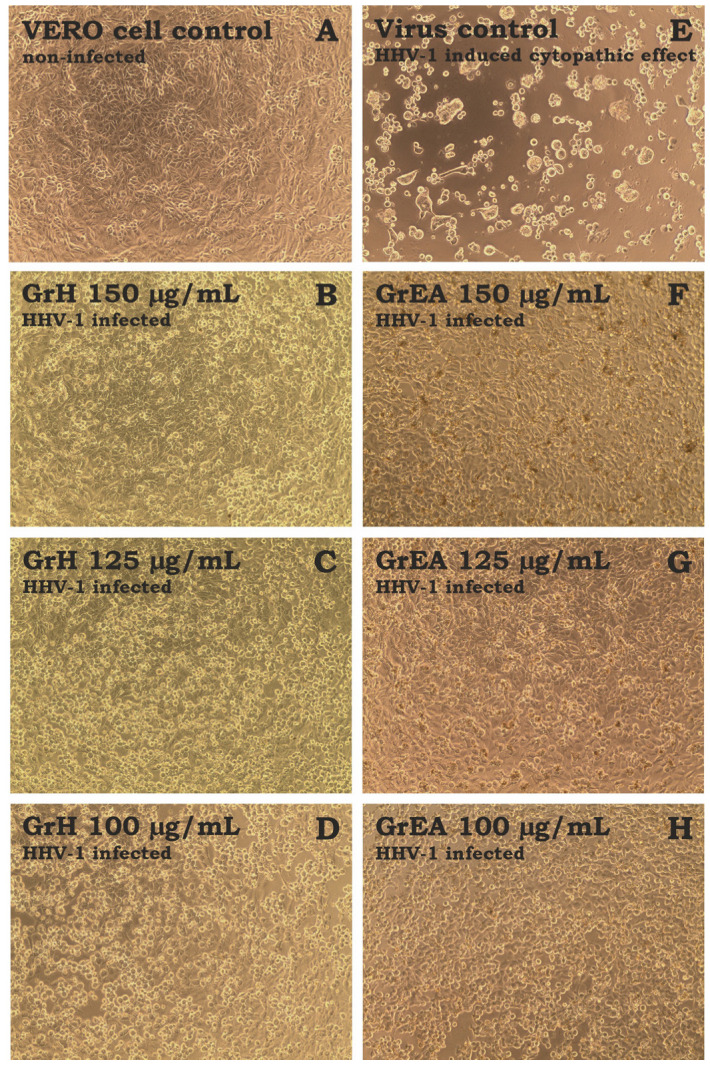
The influence of *G. robertianum* extracts on the development of CPE in HHV-1-infected VERO cells. (**A**) VERO cell control; HHV-1 infected VERO cells treated with GrH 150 µg/mL (**B**), GrH 125 µg/mL (**C**), GrH 100 µg/mL (**D**), GrEA 150 µg/mL (**F**), GrEA 125 µg/mL (**G**), GrEA 100 µg/mL (**H**); (**E**) Virus control, HHV-1 infected and non-treated VERO cells.

**Figure 5 pharmaceutics-15-01561-f005:**
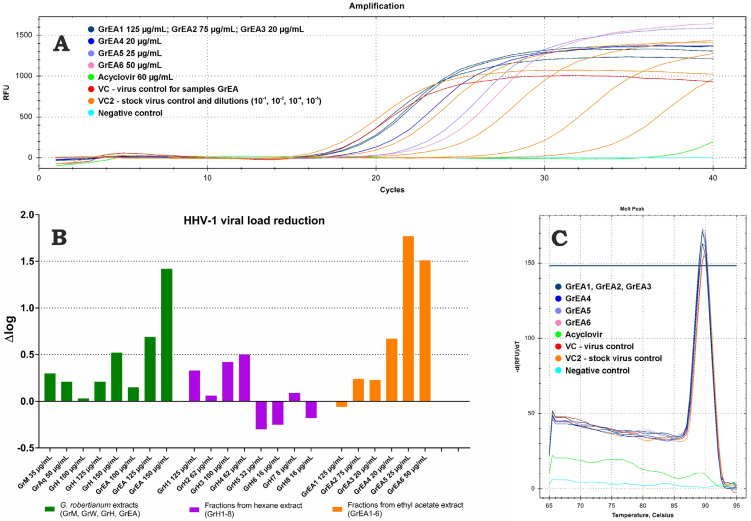
Evaluation of the HHV-1 viral load using real-time PCR. ((**A**) Real-time PCR amplification curve; (**B**) reduction of HHV-1 viral load by *G. robertianum* extracts and fractions; (**C**) melt curve analysis).

**Figure 6 pharmaceutics-15-01561-f006:**
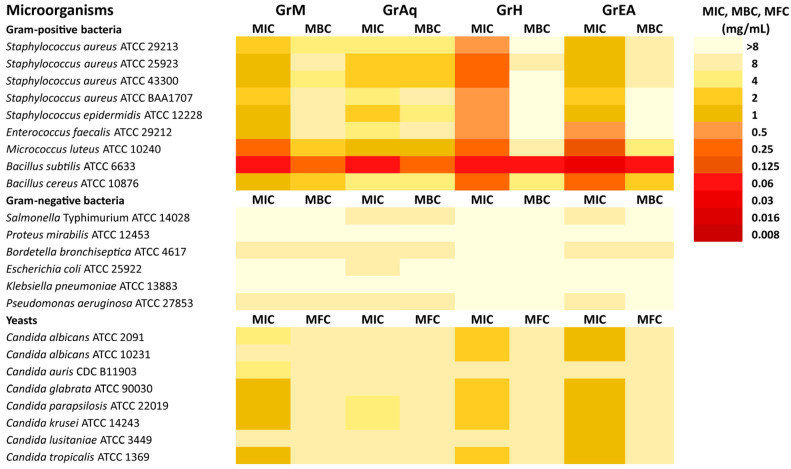
Antimicrobial activity of *G. robertianum* extracts (MIC—minimum inhibitory concentration (mg/mL); MBC—minimum bactericidal concentration (mg/mL); MFC—minimum fungicidal concentration (mg/mL); reference antimicrobial substances MIC values: fluconazole 1 μg/mL for *Candida albicans* ATCC 10231, vancomycin 1 μg/mL for *Staphylococcus aureus* ATCC 29213, and ciprofloxacin 0.015 μg/mL for *Escherichia coli* ATCC 25922).

**Table 1 pharmaceutics-15-01561-t001:** Evaluation of cytotoxicity of *G. robertianum* extracts.

Extract	VERO	FaDu		Detroit 562		RKO	
	CC_50_	CC_50_	SI	CC_50_	SI	CC_50_	SI
Methanolic	187.17 ± 28.32	115.9 ± 11.33	1.61	188.65 ± 10.68	0.99	180.57 ± 16.80	1.04
Aqueous	313.17 ± 35.3	145.83 ± 13.46	2.15	381.1 ± 8.49	0.82	284.93 ± 10.15	1.1
Hexane	286.97 ± 29.39	109.23 ± 8.62	2.63	117.57 ± 9.43	2.44	98.91 ± 3.61	2.9
Ethyl acetate	278.13 ± 19.08	63.36 ± 8.13	4.39	137.51 ± 9.88	2.02	63.63 ± 8.13	4.37

CC_50_—50% cytotoxic concentration (µg/mL); mean ± SD; SI—selectivity index (SI = CC_50_VERO/CC_50_CancerCells); Cell lines: VERO—green monkey kidney; FaDu—human hypopharyngeal squamous cell carcinoma; Detroit 562—human pharyngeal carcinoma; RKO—human colon cancer.

**Table 2 pharmaceutics-15-01561-t002:** Cytotoxicity of fractions from *G. robertianum* hexane and ethyl acetate extracts.

Fractions	VERO	FaDu		Detroit 562		RKO	
	CC_50_	CC_50_	SI	CC_50_	SI	CC_50_	SI
GrH1	330.05 ± 26.23	203.95 ± 23.41	1.62	273.65 ± 20.58	1.21	238.5 ± 22.97	1.38
GrH2	302.5 ± 3.25	142.55 ± 22.27	2.12	224.85 ± 20.44	1.35	168.62 ± 11.39	1.79
GrH3	288.2 ± 5.94	101.9 ± 1.13	2.83	102.69 ± 4.55	2.81	99.52 ± 6.68	2.90
GrH4	180.15 ± 6.72	67.62 ± 2.84	2.66	129.85 ± 6.86	1.39	89.68 ± 4.52	2.01
GrH5	113.3 ± 8.49	46.25 ± 3.92	2.45	88.71 ± 4.74	1.28	49.16 ± 1.46	2.30
GrH6	70.88 ± 3.77	22.04 ± 1.82	3.22	48.71 ± 1.36	1.46	29.59 ± 0.99	2.40
GrH7	30.82 ± 0.31	13.35 ± 1.11	2.31	31.87 ± 0.25	0.97	187.0 ± 14.9	0.16
GrH8	34.15 ± 0.14	18.51 ± 2.52	1.85	34.61 ± 0.32	0.99	125.10 ± 9.43	0.27
GrEA1	289.07 ± 16.05	82.69 ± 1.94	3.50	168.80 ± 7.5	1.71	89.67 ± 5.42	3.22
GrEA2	147.87 ± 4.66	49.02 ± 4.54	3.02	93.78 ± 5.02	1.58	65.6 ± 1.07	2.25
GrEA3	65.01 ± 0.62	19.86 ± 0.73	3.27	45.06 ± 6.05	1.44	26.73 ± 1.28	2.43
GrEA4	43.83 ± 3.23	14.22 ± 1.43	3.08	27.98 ± 6.50	1.57	22.55 ± 0.61	1.94
GrEA5	43.46 ± 0.29	17.89 ± 0.68	2.43	40.23 ± 3.44	1.08	39.28 ± 2.09	1.11
GrEA6	67.29 ± 3.54	35.48 ± 4.89	1.90	61.55 ± 6.33	1.09	51.86 ± 1.15	1.30

CC_50_—50% cytotoxic concentration (µg/mL); mean ± SD; SI—selectivity index (SI = CC_50_VERO/CC_50_CancerCells); GrH1–GrH8—fractions 1–8 from hexane extract; GrEA1–GrEA6—fractions 1–6 from ethyl acetate extract; cell lines: VERO—green monkey kidney; FaDu—human hypopharyngeal squamous cell carcinoma; Detroit 562—human pharyngeal carcinoma; RKO—human colon cancer.

## Data Availability

All data generated within this research were included in the manuscript.

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
