# Peer review of "Herb Robert’s Gift against Human Diseases: Anticancer and Antimicrobial Activity of Geranium robertianum L."

_pharmaceutics, 2023, doi:10.3390/pharmaceutics15051561_

Round 1

Reviewer 1 Report

The study of G. robertianum done by Świątek et al arouses general interest and adds to the existing knowledge. However, the organization of the research questions and subsequent studies has made some irrelevant sections in the manuscript making it a bit tiresome to read. It is suggested that after the authors change the writing of the manuscript, it should be extensively copyedited for better readability.

Major Comments:

·         Identification of G. robertianum should be made from a local accredited herbarium and an identification/accession number should be added to the Plant Material section. This is mandatory.

·         The entire Introduction section needs to be rewritten, mostly in the light of available knowledge about G. robertianum and the gaps thereof. More emphasis should be given on the rationale of the study. Studies done in the Ref. 2 of the manuscript needs to be elaborated more.

·         3rd para in Introduction is unnecessarily long with available treatments for HHV-1 and possible mechanisms thereof. This is not required since the authors are not elucidating any pathways for the activity of G. robertianum. Please shorten it in the context of the current study to make it more comprehensible.

·         What is the difference between “2.5.1. Evaluation of cytotoxicity” in Methods and “The cytotoxicity testing” in the Supplementary? Please reduce this unnecessary lengthening of the manuscript. Also, kindly provide cell number, conc. range of the extract(s) used, etc. to allow reproducibility, rather than technical details like names of the instruments.

·         The entire Table 1 should be given in the Supplementary.

·         The authors have interchangeably used “cytotoxic” and “antineoplastic” to describe the potential activity of the plant extracts (Ref. doi:10.1186/s40199-015-0102-0). However, it is not possible to determine whether a material is cytotoxic (toxic for cells) or antineoplastic (growth/proliferation inhibitory for malignant cells) by only MTT assay; and it can only be assessed at least by 3H-thymidine, BrdU, cell cycle analysis, etc. Hence, it is suggested that the authors refrain from using the word antineoplastic (may use anticancer) and rewrite the sections where it is mentioned. Also, please don’t use the words simultaneously, e.g., cytotoxicity and antineoplastic potential, Cytotoxicity and anticancer properties, etc.

·         Figures 4 and 5 can be given in the Supplementary, especially after removing the non-essential (in which fractions remain inactive) photomicrographs.

·         Where are the results for control in Figures 7, 8, and 9? Besides the most important results, most gram-neg bacteria and yeast results can be moved to the Supplementary. The figures and associated legends are to be modified as well.

·         The lengthy Discussion section is written as an extension of the Results section and contains little explanation of the results obtained. The authors should rewrite this section focusing on explaining the current observation in the light of available literature. Since no mechanism has been shown for the extract(s) activities depicted in the manuscript, it should be very to the point avoiding unnecessary speculation. And above all, the length of the Discussion should be reduced as much as plausible.

Minor Comments:

·         What is “ellagotannins”? Please explain since no reference for that word is found.

·         The authors have not performed anti-diabetic and/or antioxidant/free radical scavenging studies. So, what is the rationale of the 2nd para of the Introduction?

·         MTT is not an acronym for microculture tetrazolium assay. Kindly use MTT (3-(4, 5-dimethylthiazolyl-2)-2, 5-diphenyltetrazolium bromide) or simply MTT assay.

·         Table 3 should come after Figures 2 and 3 to ensure continuity.

·         Please elaborate each section of Figure 6 in the legend. And please put redundant sections in the Supplementary.

·         Please do not use specific words or terms at will. E.g., fluconazole, ciprofloxacin & vancomycin are not “chemotherapeutic” agents (lines 237-8).

· 

Overall copyediting of the manuscript from a professional service is highly recommended.

Author Response

The study of G. robertianum done by Świątek et al arouses general interest and adds to the existing knowledge. However, the organization of the research questions and subsequent studies has made some irrelevant sections in the manuscript making it a bit tiresome to read. It is suggested that after the authors change the writing of the manuscript, it should be extensively copyedited for better readability.

Answer: Dear Reviewer, thank you for your expert insight into our work. We’ve done our best to modify the manuscript following your comments. Hopefully, you find our work satisfactory.

Major Comments:

Identification of G. robertianum should be made from a local accredited herbarium and an identification/accession number should be added to the Plant Material section. This is mandatory.

Answer: Dear Reviewer, we acknowledge your concern about the plant material identification and its quality. This is also an important issue for us, which is why the dried plant material used was a commercially available product manufactured by DARY NATURY sp. z o.o. Koryciny 73, 17-315 Grodzisk, Poland. This is a certified manufacturer and distributor of herbal products. Each batch of plant material undergoes identification, quality control, and certification. The herb of G. robertianum was cultivated in organic farming, under the European organic trademark (EKO-trademark). All products of this company are certified as organic products (Certificate PL-EKO-01-001493) according to the Regulation (EU) 2018/848 of the European Parliament and of the Council of 30 May 2018 on organic production and labeling of organic products. Appropriate information was included in the text.

The entire Introduction section needs to be rewritten, mostly in the light of available knowledge about G. robertianum and the gaps thereof. More emphasis should be given on the rationale of the study. Studies done in the Ref. 2 of the manuscript needs to be elaborated more.

Answer: Dear Reviewer, following your suggestions, we have modified the introduction section. Information on the HHV-1 was shortened, as well as the treatment of herpes. We have elaborated more on Ref. 2 (Graça, V. C. et al.), by providing more detailed information on phytochemical and biological studies of this plant. Also, we have emphasised the rationale of our study. Thank you for your valuable insight which allowed us to improve the Introduction.

3rd para in Introduction is unnecessarily long with available treatments for HHV-1 and possible mechanisms thereof. This is not required since the authors are not elucidating any pathways for the activity of G. robertianum. Please shorten it in the context of the current study to make it more comprehensible.

Answer: Dear Reviewer, according to your suggestions, the 3rd paragraph of the Introduction was shortened, and the information on anti-herpesviral treatment was reduced. 

What is the difference between “2.5.1. Evaluation of cytotoxicity” in Methods and “The cytotoxicity testing” in the Supplementary? Please reduce this unnecessary lengthening of the manuscript. Also, kindly provide cell number, conc. range of the extract(s) used, etc. to allow reproducibility, rather than technical details like names of the instruments.

Answer: Dear Reviewer, thank you for this suggestion. We have reduced the 2.5.1 section accordingly, concentration ranges were included. Cell densities used during passages were included in the updated Supplementary Materials (S-M). Also, to reduce the length, the cycling parameters were removed from section 2.5.2, but are still present in the S-M.

The entire Table 1 should be given in the Supplementary.

Answer: Dear Reviewer, according to your suggestion, Table 1 was moved to the Supplementary.

The authors have interchangeably used “cytotoxic” and “antineoplastic” to describe the potential activity of the plant extracts (Ref. doi:10.1186/s40199-015-0102-0). However, it is not possible to determine whether a material is cytotoxic (toxic for cells) or antineoplastic (growth/proliferation inhibitory for malignant cells) by only MTT assay; and it can only be assessed at least by 3H-thymidine, BrdU, cell cycle analysis, etc. Hence, it is suggested that the authors refrain from using the word antineoplastic (may use anticancer) and rewrite the sections where it is mentioned. Also, please don’t use the words simultaneously, e.g., cytotoxicity and antineoplastic potential, Cytotoxicity and anticancer properties, etc.

Answer: Dear Reviewer, thank you for your watchfulness. According to your suggestions, we have refrained from using the term “antineoplastic”. Appropriate sections of the manuscript were rewritten.

Figures 4 and 5 can be given in the Supplementary, especially after removing the non-essential (in which fractions remain inactive) photomicrographs.

Answer: Figure 4 was modified and non-active fractions were removed, according to the Reviewers’ comments. In Figure 4 we have retained results for lower concentrations of GrH and GrEA, since they are relevant to show the dose-response effect of those extracts on the CPE. Figure 5 has been moved to the Supplementary materials.

Where are the results for control in Figures 7, 8, and 9? Besides the most important results, most gram-neg bacteria and yeast results can be moved to the Supplementary. The figures and associated legends are to be modified as well.

Answer: Dear Reviewer, the results for controls were added to the Figure 6 (Previously Figure 7) caption, also they can be found in the main text, lines 741 – 743. We did not want to divide the results from Figures 8 and 9 between the main text and Supplementary Materials, hence we moved them to the Supplementary files (Figures S4 and S5).

In addition, standard controls are not performed for all reference strains of both Gram-positive and Gram-negative bacteria and yeasts of the genus Candida. This is what EUCAST’s guidelines and research are for: https://mic.eucast.org/
On this page, you can check the MIC range of antibiotics/chemotherapeutics for individual microorganisms, including bacteria and fungi. Therefore, in our study, controls were performed only for representatives of each group of microorganisms: vancomycin against S. aureus ATCC 29213 (gram-positive bacteria), ciprofloxacin against E. coli ATCC 25922 (gram-negative bacteria) and fluconazole for C. albicans ATCC 10231 (fungi). We do this to make sure our results correspond with EUCAST’s guidelines.

The lengthy Discussion section is written as an extension of the Results section and contains little explanation of the results obtained. The authors should rewrite this section focusing on explaining the current observation in the light of available literature. Since no mechanism has been shown for the extract(s) activities depicted in the manuscript, it should be very to the point avoiding unnecessary speculation. And above all, the length of the Discussion should be reduced as much as plausible.

Answer: We acknowledge the Reviewer’s suggestions. The Discussion was shortened and unnecessary information was removed.  We’ve also extended the explanation of the presented results in light of the available literature. Also, we have limited speculations as much as possible. We hope that the corrected version of the Discussion is acceptable. Once again, thank you for your valuable suggestions.

Minor Comments:

What is “ellagotannins”? Please explain since no reference for that word is found.

Answer: Dear Reviewer, thank you for your watchfulness, we meant ellagitannins, which are hydrolyzable tannins. The text was corrected.

The authors have not performed anti-diabetic and/or antioxidant/free radical scavenging studies. So, what is the rationale of the 2nd para of the Introduction?

Answer: Thank you for this valuable remark, the 2nd paragraph of the Introduction was removed.

MTT is not an acronym for microculture tetrazolium assay. Kindly use MTT (3-(4, 5-dimethylthiazolyl-2)-2, 5-diphenyltetrazolium bromide) or simply MTT assay.

Answer: Thank you for pointing this out. Corrections were applied to the text.

Table 3 should come after Figures 2 and 3 to ensure continuity.

Answer: According to your suggestions, Table 3 (Cytotoxicity of fractions from G. robertianum hexane and ethyl acetate extracts) has been moved and can now be found after Figures 2 and 3 (Now it’s Table 2).

Please elaborate each section of Figure 6 in the legend. And please put redundant sections in the Supplementary.

Answer: Figure 6 was corrected (now it’s Figure 5), and three sections were excluded and moved to the supplementary file. Description of each section was provided.

Please do not use specific words or terms at will. E.g., fluconazole, ciprofloxacin & vancomycin are not “chemotherapeutic” agents (lines 237-8).

Answer: Dear Reviewer, following your suggestions, we have removed the phrase “chemotherapeutic”. We understand that the phrase “chemotherapy” is commonly associated with cancer. However, we would like to point out that fluconazole, ciprofloxacin and vancomycin are used in antimicrobial chemotherapy. The term antibacterial chemotherapy is commonly used in scientific literature and in practice, please see the “British Society for Antimicrobial Chemotherapy” (https://bsac.org.uk/) or the journal “Antimicrobial Agents and Chemotherapy” (https://journals.asm.org/journal/aac)

Reviewer 2 Report

Dear Ms. Jamila Wang,

the paper is well written and the

experiments well designed.

Please check the formula reported in the table.

It is reviewer opinion that the

paper can be accepted after few

modification.

Best Regards

Author Response

the paper is well written and the experiments well designed.

Please check the formula reported in the table.

It is reviewer opinion that the paper can be accepted after few modification.

Answer: Dear Reviewer, thank you for your kind words. The manuscript was improved. Also, one error in Table S3 was corrected.

Reviewer 3 Report

The aim of the study was to evaluate the antimicrobial and antitumor potential of Geranium robertianum extract. The study seems complex but needs some clarification, namely: 

Who identified the plant and where was the voucher specimens stored? The company from which you purchased it? Where did the plant come from? Wild or cultivated?

For cytotoxicity assessment which of the extracts you used is not specified at all. Only in the results do you indicate the four types of extracts. 

In the supplementary material you have indicated that you also used the HeLa line which is not mentioned in the manuscript. 

Why did you use 2% foetal serum for propagation? 

What were the concentrations of extract tested for cytotoxicity?

There are many things missing from the materials and methods that only appear in the results. For the study to be credible and reproducible this information must necessarily appear in the material and methods. 

What did you use as positive and negative controls for cytotoxicity? Did you also test the solvents used for extraction (hexane, methyl acetate, methanol and water)?

Do you consider hexane, methyl acetate and methanol extracts to be suitable for use in therapy? Are they not potentially toxic? 

After dissolving the formazan crystals why did you leave at 37 degrees for 24 hours? Why wasn't the chromogenic reaction evaluated after dissolution?

How did you calculate the CC50 (50% cytotoxic concentration)??

The added pictures with cells are not very clear can you enlarge a bit to be assessable? 

For the evaluation of the antiviral potential why did you use as stated in the supplementary material - The antiviral activity of propolis and honey extracts was tested against HHV-1 (ATCC, Cat. No. VR-260) propagated in the VERO cell line. ???

I think this is another experiment.

To evaluate the antimicrobial potential of the bacterial strains why were they kept at 35 degrees? What is the explanation that the extracts used have better antimicrobial potential in gram positive bacteria? Can you discuss a bit about the mechanisms involved in this possible effect besides those described in the discussion?  And the same for Candida spp. strains where no possible mechanisms are mentioned? Can you explain, obviously correlated with the literature. 

In the discussion you have to correlate the chemical composition of the extracts with the biological effect. You should also consider the possible adverse effect that may occur by using these solvents in the extraction. 

Harmonisation of the data in the article with that in the supplementary material is necessary. 

Author Response

The aim of the study was to evaluate the antimicrobial and antitumor potential of Geranium robertianum extract. The study seems complex but needs some clarification, namely: 

Answer: Dear Reviewer, thank you for your expertise. We’ve done our best to address all issues and questions.

Who identified the plant and where was the voucher specimens stored? The company from which you purchased it? Where did the plant come from? Wild or cultivated?

Answer: Dear Reviewer, we acknowledge your concern about the plant material identification and its quality. The dried plant material used in this study was a commercially available product manufactured by DARY NATURY sp. z o.o. Koryciny 73, 17-315 Grodzisk, Poland. This is a certified manufacturer and distributor of herbal products. Each batch of plant material undergoes identification, quality control, and certification. The herb of G. robertianum was cultivated in organic farming, under the European organic trademark (EKO-trademark). All products of this company are certified as organic products (Certificate PL-EKO-01-001493) according to the Regulation (EU) 2018/848 of the European Parliament and of the Council of 30 May 2018 on organic production and labeling of organic products. Appropriate information was included in the text.

For cytotoxicity assessment which of the extracts you used is not specified at all. Only in the results do you indicate the four types of extracts. 

Answer: Dear Reviewer, thank you for your watchfulness, information was added to the text – “The cytotoxicity of crude extracts (aqueous, methanolic, ethyl acetate and hexane), as well as fractions obtained from ethyl acetate extract (GrEA1-GrEA6) and hexane extract (GrH1-GrH8) was evaluated using a previously described [5] MTT assay”

In the supplementary material you have indicated that you also used the HeLa line which is not mentioned in the manuscript. 

Answer: Dear Reviewer, we are sorry for this mistake. The HeLa cells were not used in this particular research. Thank you very much for pointing this out. The supplementary materials were corrected.

Why did you use 2% foetal serum for propagation? 

Answer: Dear Reviewer, to clarify, 10% of fetal bovine serum (FBS) was used for cell passaging to stimulate cellular divisions and obtain semiconfluent monolayers. Whereas 2% FBS was used to maintain cells after passages and also for all experiments. Information can be found in the supplementary materials. This is a standard practice in cell culturing since cells require hormones and other growth factors which are found in FBS. Of course, control cells in all experiments are always supplemented with media containing the same amount of FBS to exclude its influence on the experimental procedure. 

What were the concentrations of extract tested for cytotoxicity?

Answer: Dear Reviewer, the concentration ranges are now included in the text – “Briefly, the cellular monolayers of appropriate cell lines in 96-well flat-bottomed plates were treated with serial dilutions of stock extracts (1000 – 1 µg/mL) or fractions (500 – 0.5 µg/mL) in cell media for 72 h.”

There are many things missing from the materials and methods that only appear in the results. For the study to be credible and reproducible this information must necessarily appear in the material and methods. 

Answer: Dear Reviewer, we have updated the information in Materials and Methods. However, one of the Reviewers asked us to shorten the manuscript, including the Materials and Methods, hence we had to move some information to the Supplementary Materials.

What did you use as positive and negative controls for cytotoxicity? Did you also test the solvents used for extraction (hexane, methyl acetate, methanol and water)?

Answer: Dear Reviewer, the solvents used for extractions do not need to be tested since the solvents are removed under reduced pressure until dryness from the extract after the extraction, and then the dry residue is dissolved in DMSO (the stock solutions concentration was 50 mg/mL). The cytotoxicity of DMSO in the concentrations equal to those present in the dilutions of stock solutions were tested to exclude any effect on the assay (this information is now included in the Supplementary Materials). There are no approved substances for negative control of cytotoxicity. For our studies, we use cells supplemented with complete cell media as a negative control (cell control). In this particular study, we have not used a positive control because we were conducting a screening of plant extract activity. Various studies use cytotoxic compounds or anticancer drugs, like cisplatin, doxorubicin, etoposide, or hydroxycarbamide, as a positive control during in vitro studies of cytotoxicity and anticancer properties. However, we believe that using those substances as a positive control in the studies of plant extract is not always necessary. Plant extracts are complex mixtures of various molecules, and it’s difficult to compare their activity with pure compounds. Of course, if particular compounds are isolated from plant extract and show significant anticancer potential, their activity should be compared to standard anticancer drugs.

Do you consider hexane, methyl acetate and methanol extracts to be suitable for use in therapy? Are they not potentially toxic? 

Answer: Dear Reviewer, as we have mentioned above, the solvents were evaporated under reduced pressure until dryness. That is why the solvents used for extraction are not relevant as far as potential toxicity is concerned. Of course, it is possible that different extracts may contain potentially harmful substances, but this is not the result of the use of a particular solvent.

After dissolving the formazan crystals why did you leave at 37 degrees for 24 hours? Why wasn't the chromogenic reaction evaluated after dissolution?

Answer: the formazan crystals require several hours to dissolve in our solvent (SDS/DMF/PBS). From our experience, we know that overnight incubation is an adequate approach. Also, this is done similarly to a paper by Takenouchi T. and Munekata E. which is commonly cited as an example of MTT procedure.

Takenouchi, T., & Munekata, E. (1998). Amyloid β-Peptide-induced Inhibition of MTT Reduction in PC12h and C1300 Neuroblastoma Cells: Effect of Nitroprusside. Peptides, 19(2), 365–372. doi:10.1016/s0196-9781(97)00377-x 

How did you calculate the CC50 (50% cytotoxic concentration)??

Answer: Dear Reviewer, the Synergy H1 Multi-Mode Microplate Reader with Gen5 software (ver. 3.09.07; BioTek Instruments, Inc.) was used to measure the absorbance (540 and 620 nm). The results were blanked (ODsample-ODblank, blank – medium without extract), then delta values (blankedOD540 nm – blankedOD620 nm, OD-optical density) are calculated, and then normalization (percentage of viability in relation to the mean viability of control cells) is performed automatically by the Gen5 software (we have optimized protocols for this). Then the results (percentage of viability at different dilutions) are exported to and evaluated using GraphPad Prism (version 7.04, GraphPad Software). The CC50 (concentration decreasing the viability by 50%) values were calculated from dose-response curves (non-linear regression model). These are non-linear dose-response equations included in the GraphPad Prism.

The added pictures with cells are not very clear can you enlarge a bit to be assessable? 

Answer: Dear Reviewer, some of the figures were moved from the main text as requested by other Reviewer. We will do our best to modify the quality of the ones remaining.

For the evaluation of the antiviral potential why did you use as stated in the supplementary material - The antiviral activity of propolis and honey extracts was tested against HHV-1 (ATCC, Cat. No. VR-260) propagated in the VERO cell line. ??? I think this is another experiment.

Answer: Dear Reviewer, once again, we are sorry for this mistake. As we have stated above, we were working on another manuscript at the same time, which is about propolis and honey extracts, and some information got mixed up in the supplementary files. Supplementary material was corrected.

To evaluate the antimicrobial potential of the bacterial strains why were they kept at 35 degrees?

Answer: Dear Reviewer, the evaluation of antimicrobial potential was done according to the European Committee on Antimicrobial Susceptibility Testing (EUCAST), where 35±1ºC temperature is required (MIC determination – broth microdilution according to ISO standard 20776-1)

What is the explanation that the extracts used have better antimicrobial potential in gram positive bacteria? Can you discuss a bit about the mechanisms involved in this possible effect besides those described in the discussion?  And the same for Candida spp. strains where no possible mechanisms are mentioned? Can you explain, obviously correlated with the literature. 

Answer: Dear Reviewer, thank you for this comment. The difference between G-positive and Gram-negative bacteria may be due to the different structures of the cell walls, as we have stated in the last section of our Discussion. Candida, as a Eukaryote, has not only different cellular morphology but also different metabolism, hence it reacts to xenobiotics in a different way.  Of course, this was only a speculation. Without evaluating the actual mechanism of action it’s difficult to discuss those differences. One of the Reviewers asked us to avoid speculating and modify the Discussion. We’ve made corrections to the manuscript and hope you find them appropriate.

In the discussion you have to correlate the chemical composition of the extracts with the biological effect. You should also consider the possible adverse effect that may occur by using these solvents in the extraction. 

Answer: Dear Reviewer, we have modified the Discussion according to your suggestions. Compounds with previously reported anticancer or antiviral activities were mentioned where applicable. Concerning the solvents, as we have previously mentioned, they were removed from the extracts, hence they pose no cytotoxicity risk.

Harmonisation of the data in the article with that in the supplementary material is necessary. 

Answer: Dear Reviewer, we have re-checked and harmonized the Supplementary Materials with the manuscript text.

Round 2

Reviewer 1 Report

I think the manuscript has been much revised and upgraded to be accepted, should it satisfies all the journal norms.

Reviewer 3 Report

The corrections indicated have largely been made. The paper in its present form can be considered suitable for publication